# Calibration, Entropy Rates, and Memory in Language Models

## Abstract

Building accurate language models that capture meaningful long-term dependencies is a core challenge in natural language processing. Towards this end, we present a calibration-based approach to measure long-term discrepancies between a generative sequence model and the true distribution, and use these discrepancies to improve the model. Empirically, we show that state-of-the-art language models, including LSTMs and Transformers, are *miscalibrated*: the entropy rates of their generations drift dramatically upward over time. We then provide provable methods to mitigate this phenomenon. Furthermore, we show how this calibration-based approach can also be used to measure the amount of memory that language models use for prediction.

## 1 Introduction

Recent advances in language modeling have resulted in significant improvements on a wide variety of benchmarks (Dai et al., 2018; Gong et al., 2018; Takase et al., 2018). Capturing long-term dependencies has especially been a major focus, with approaches ranging from explicit memory-based neural networks (Grave et al., 2016; Ke et al., 2018) to optimization improvements to stabilize learning (Le et al., 2015; Trinh et al., 2018). However, while these techniques seem to improve on standard metrics like perplexity and even produce remarkably coherent text (Radford et al., 2019), we still do not have appropriate measures to assess long-term properties in language models, making it difficult to choose between different model options for downstream tasks.

Starting from Shannon's seminal work that essentially introduced statistical language modeling (Shannon, 1951), the most classical and widely studied long-term property of a language model is its entropy rate — the average amount of information contained per word, conditioned on the preceding words. A learned model provides an upper bound for the entropy rate of a language, via its cross-entropy loss. The exponential of the entropy rate can be interpreted as the effective support size of the distribution of the next word (intuitively, the average number of "plausible" word choices to continue a document), and the perplexity score of a model (the exponential of the cross entropy loss) is an upper bound for this quantity. In state-of-the-art models trained on billion-scale corpora, this number ranges between 10 and 30 (Melis et al., 2017; Radford et al., 2019). A natural diagnostic question, with which we begin our work, is whether the long-term generations of these models exhibit the same entropy rates as the underlying languages they are modeling predictively.

| Model | Corpus | Test ppl. | $e^{\textbf{EntRate}}$ |
|---|---|---|---|
| AWD-LSTM (Merity et al., 2017) | PTB | 58.3 | 93.1 |
| CNN-LSTM (Jozefowicz et al., 2016) | GBW | 29.8 | 49.4 |
| Transformer (Vaswani et al., 2017b) | GBW | 28.1 | 34.7 |
| GPT-2 (Radford et al., 2019) | WebText | 23.7 | 61.2 |

Table 1: Perplexity degradations for generations from popular language models. State-of-the-art performance is usually reported via perplexity with respect to the test corpus (one-step prediction loss), but there is a striking blowup in the perplexity (i.e. exponential of the entropy) of these models' long-term generations. **Test ppl.** is the exponential of the *cross-entropy* of the model with respect to the test corpus.

Empirically, and perhaps surprisingly, it turns out that the entropy rate of generated text is *substantially* higher than the estimate for true text derived from the model's one-step predictions. As seen in Table 1 (see also Figure 1), this is true for both state-of-the-art LSTMs and Transformers trained on a variety of datasets. As a timely example, the GPT-2 model (Radford et al., 2019), the object of much recent attention for its seemingly coherent and on-topic generations, suffers a dramatic degradation in its entropy rate, from 23.7 to 61.2.

This empirical finding is notable since the neural attention- and memory-based techniques have been steadily improving on standard metrics like perplexity and, in some cases, even produce remarkably coherent text (often with some heuristics to reject poor generations). That the perplexity of generated text is so much higher than it is under the true distribution suggests that there are significant gaps in our current methodologies in accurately learning language models, particularly if we are interested in generating text that globally resembles the modeled language itself.

**Our contributions.** The focus of this work is twofold: to improve generations based on any measurement mismatch on a long-term property of the model (e.g. the entropy rate), and to quantify the way a model's predictions depend on the distant past. Central to both of these is a calibration-based approach, which is utilized in statistics and other areas of machine learning (Dawid, 1982; 1985; Foster, 1991; Zadrozny and Elkan, 2002; Platt, 1999; Guo et al., 2017; Niculescu-Mizil and Caruana, 2005).

First, we show that, from a worst-case perspective, even an extremely accurate model (with $\varepsilon$ average KL divergence from the true distribution) may have generated text with a substantially different entropy rate as compared to the true distribution. Indeed, we show that this worst-case amplification may occur for a variety of long-term properties of a probabilistic language model; this is because the one-step KL divergence does not in general provide tight control over the expectation of a bounded function. The observed entropy rate amplification (as seen in Table 1) demonstrates that this is not only of theoretical concern. We then describe a calibration procedure to fix this mismatch while simultaneously improving the perplexity of the language model. From a statistical perspective, the procedure is simple, and we discuss approaches to make it computationally efficient.

Second, we provide a definition for long-term memory in language models as the mutual information between the models predictions and the distant past in the input. We then provide an upper bound on the amount of this mutual information using calibrated distributions (with a single-parameter exponent). This allows us to estimate the amount of context used by a language model as a function of the distance of past tokens from the current prediction time step.

We perform empirical studies to accompany our theoretical results. We first use the entropy rate calibration algorithm to fix an LSTM language model, resulting in a drop of around 20 perplexity points in the generated text (so that the entropy rate of the model more accurately matches that of the language itself). Then, we empirically estimate and compare the long-term memory of state-of-the-art language models. Our insights point towards new ways of assessing (and fixing) language models, especially in terms of their long-term properties, in a manner complementary to existing metrics like perplexity.

## 2 RELATED WORK

**Improving language modeling with long-term dependencies.** Recent approaches to improving language modeling have focused on several ways to better capture long-term dependencies, from using manually-defined context representations (Mikolov and Zweig, 2012; Ji et al., 2015; Wang and Cho, 2016) or document-level topics (Wang et al., 2017) to using LSTM recurrent neural networks with careful initialization (Le et al., 2015), auxiliary loss signals (Trinh et al., 2018) or augmented memory structures (Grave et al., 2016; Ke et al., 2018). Wiseman and Rush (2016) use scoring functions over *sequences* and search-based optimization to improve generation in seq2seq models.

More recent work has demonstrated the applicability of Transformer networks (Vaswani et al., 2017a) to the task, potentially side-stepping issues in training recurrent networks (e.g. vanishing/exploding gradients) and scaling to longer contexts (Dai et al., 2018; Radford et al., 2018). All these papers propose either architectural or optimization innovations to improve language model

training. In contrast, we define and measure explicit long-term properties of language models and show that calibrating them correctly can provide improvements to any black-box language model.

Recent empirical breakthroughs have stemmed from language models which do not specify a unique autoregressive factorization (Devlin et al., 2018; Yang et al., 2019; Liu et al., 2019), and thus do not specify a unique $\widehat{Pr}$. It remains an interesting problem to identify and sample from distributions induced by these models (Wang and Cho, 2019); thus, our end-to-end theoretical guarantees do not hold in this setting.

**Information-theoretic approaches.** While most language models aim to predict a distribution over the next token conditioned on the context, there have been alternative approaches relying on information-theoretic measures. Jost and Atwell (1994) propose a model which makes use of mutual information between word pairs to generate word sequences that retain longer-term dependencies. McAllester (2018) propose a training objective based on mutual information for predictive modeling, and demonstrate its application for phoneme prediction. Clarkson and Robinson (1999) develop a hybrid metric using both perplexity and entropy rate, and show that it correlates better with a downstream metric like word error rate. Such works propose alternative optimization objectives; in contrast, we show how to use information-theoretic measures to improve models with respect to existing objectives like cross-entropy.

**Measuring long-term statistics.** Khandelwal et al. (2018) analyze LSTM-based language models and empirically show that such models make use of a finite context for prediction. Lin and Tegmark (2017) measure mutual information between any two symbols in human languages, and show that it decays with distance, roughly following a power law distribution. Takahashi and Tanaka-Ishii (2018) provide an upper bound for the entropy (character-level) of human languages by training neural language models with various context and data sizes and extrapolating to infinity. While we also make use of measures like entropy and mutual information across longer contexts, our goal is to use these to better calibrate the language model and provably improve its perplexity.

**Calibration and integral probability metrics.** The idea of matching properties of the models' predictions to the empirical outcomes, in an online setting, goes back (at least) to the "prequential principle" of Dawid (1982; 1985), with subsequent work in online and game-theoretic settings (Foster, 1991; Vovk, 2001; Kalai et al., 1999). The idea of improving probability scores is also common in machine learning (Zadrozny and Elkan, 2002; Platt, 1999; Guo et al., 2017; Niculescu-Mizil and Caruana, 2005). Recently, Ott et al. (2018) assessed model calibration for machine translation systems using word-level probabilities. The notion of examining the expectation of functions as a metric for the distance between two distributions sometimes goes under the name of integral probability metrics (Mller, 1997; Sriperumbudur et al., 2009), and this notion is becoming increasingly relevant again in unsupervised learning through the connections to GANs (Mroueh and Sercu, 2017). In this work, we directly focus on the KL divergence, where our use of calibration is largely based on basic facts about exponential families (Brown, 1986).

**Relation to generation-improving heuristics.** If the sole objective is to improve the qualitative coherency of sampled generations, a wide variety of heuristics exist in the literature. The simplest of these is a constant multiplicative adjustment to the model's logits (known as *softmax temperature* Xie (2017)). This is a specific version of our method (Algorithm 2) with a constant logistic regression feature instead of the next-token conditional entropy. Relatedly, *greedy* and *top-k* sampling (used in state-of-the-art works such as (Radford et al., 2019)) are heuristics which make local modifications to the model's conditional probabilities to decrease diversity and eliminate nonsensical generations.

Efforts to push the empirical state of the art in generation quality have given rise to more complex heuristics. Bengio et al. (2015) propose retraining the network on its own generations with a carefully scheduled probability for each token. Some works regularize a model's generations with an auxiliary reverse language model Zhang et al. (2019); Liu et al. (2016). Yet others promote realism using adversarial training protocols (Bahdanau et al., 2016; Lin et al., 2017; Fedus et al., 2018).

We stress that our calibration methods result in a provable improvement in the original training objective (i.e. lower perplexity). As far as we know, none of the aforementioned heuristic methods can hope to provide such a strong guarantee, since they are fundamentally designed to bias

models towards a different objective. Our work mitigates model hallucinations for (almost) free[1], in the sense that the global objective (entropy rate drift) is improved without worsening the local objective (perplexity). Furthermore, calibration preserves the computational efficiency of density estimation: a conditional probability vector from Algorithm 2 can be computed using $O$(vocabulary size) inferences on the original model. In the more advanced heuristics, the implied distribution over sequences is lost, and is only accessible by black-box sampling.

## 3 PRELIMINARIES

We first define some useful quantities for our analyses. Let $\Pr(W_1, W_2, \ldots, W_T)$ represent the true underlying distribution over $T$ length sequences of words, where the vocabulary is of size $M$. Let $W_{1:T}$ denote a random sequence of length $T$, with distribution $\Pr(W_{1:T})$. For clarity of exposition, we assume that all sequences (i.e. sentences or documents or books) are of equal length $T$.

For any distributions $\mathcal{D}$ and $\mathcal{D}'$ over length-$T$ sequences, recall that the entropy $H(\cdot)$, KL-divergence, and entropy rate are, respectively, defined by: $H(\mathcal{D}) := \mathbb{E}_{w_{1:T} \sim \mathcal{D}} \left[ \log \frac{1}{\mathcal{D}(W_{1:T} = w_{1:T})} \right]$,

$\mathrm{KL}(\mathcal{D} \parallel \mathcal{D}') := \mathbb{E}_{w_{1:T} \sim \mathcal{D}} \left[ \log \frac{\mathcal{D}(W_{1:T} = w_{1:T})}{\mathcal{D}'(W_{1:T} = w_{1:T})} \right]$, and $\mathrm{EntRate}(\mathcal{D}) := \frac{1}{T} H(\mathcal{D})$. Let $\widehat{\Pr}(W_{1:T})$ denote a learned distribution over sequences. In the typical sequential prediction setting, the probabilistic model is implicitly defined by the conditional distributions $\Pr(W_t | W_{<t})$, which are typically efficiently computable. It is standard for such a *language model* to be trained to minimize the *cross entropy* objective:

$$\mathrm{CE}(\Pr \parallel \widehat{\Pr}) := \frac{1}{T} \mathop{\mathbb{E}}_{w_{1:T} \sim \Pr} \left[ \sum_{t=1}^{T} \log \frac{1}{\widehat{\Pr}(w_t | w_{<t})} \right] = \frac{1}{T} \mathop{\mathbb{E}}_{w_{1:T} \sim \Pr} \left[ \log \frac{1}{\widehat{\Pr}(w_{1:T})} \right].$$

Note that for an accurate language model, we would hope that: $\mathrm{CE}(\Pr \parallel \widehat{\Pr}) \approx \mathrm{EntRate}(\widehat{\Pr})$, i.e. the entropy rate of the sequences generated under the learned model is nearly that of the cross entropy of the model (with respect to the true distribution $\Pr$).

Throughout, we assume that

$$\frac{1}{T} \mathrm{KL}(\Pr \parallel \widehat{\Pr}) = \mathrm{CE}(\Pr \parallel \widehat{\Pr}) - \mathrm{EntRate}(\Pr) \leq \varepsilon \tag{1}$$

holds for some $\varepsilon$. In other words, the (unknown) $\varepsilon$ measures the degree of sub-optimality of the learned model, this $\varepsilon$ is often referred to as the Bayes regret.

## 4 CALIBRATION AND ENTROPY RATES

In this section, we assess the long-term properties of language models when generating text. Specifically, we quantify the amplification in the entropy rate of generations under an $\varepsilon$-accurate model (Eq. 1). We then provide a procedure to fix this amplification, without increasing the perplexity of the model. Proofs for all statements are provided in the supplementary material.

For generality, consider a function $f : [M]^T \to \mathcal{R}$, defined on $T$ length sequences. Let the mean and variance of $f$ under distribution $\mathcal{D}$ be denoted by $\mu_{\mathcal{D}}(f)$ and $\sigma_{\mathcal{D}}^2(f)$

$$\mu_{\mathcal{D}}(f) := \mathop{\mathbb{E}}_{w_{1:T} \sim \mathcal{D}}[f(w_{1:T})], \quad \sigma_{\mathcal{D}}^2(f) := \mathop{\mathbb{E}}_{w_{1:T} \sim \mathcal{D}}[(f(w_{1:T}) - \mu_{\mathcal{D}}(f))^2].$$

### 4.1 ERROR AMPLIFICATION UNDER OUR MODEL

If our learned model $\widehat{\Pr}$ is accurate, we may hope that $\mu_{\Pr}(f) \approx \mu_{\widehat{\Pr}}(f)$ i.e. that the expected value of $f$ under the true distribution $\Pr$ is close to its expected value under our model. We can quantify this gap as follows:

---

[1] Technically, at the statistical cost of fitting *one* more parameter.

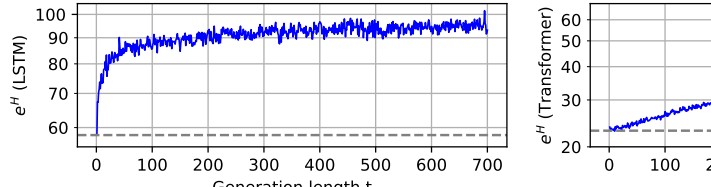

Figure 1: Perplexity (exponential of conditional entropy, given the past) of the $t$-th generated word, for two popular language models, averaged over more than 500 generation runs with different contexts. At $t = 1$, this is the model's upper bound for the language's perplexity. As $t \to \infty$, this is the exponential of the entropy rate of the model's own generations. For a perfectly calibrated model, this curve would be flat (gray dotted lines). *Left:* LSTM trained on Penn Treebank. *Right:* GPT-2 Transformer.

**Lemma 4.1.** *(Pinsker's Inequality (Csiszar and Körner, 2011)) Suppose that for all $w_{1:T}$, $f(w_{1:T}) \le B$. Then:*

$$\left|\mu_{\mathrm{Pr}}(f) - \mu_{\widehat{\mathrm{Pr}}}(f)\right| \le B\sqrt{2\mathrm{KL}(\mathrm{Pr} \parallel \widehat{\mathrm{Pr}})}.$$

Since this holds for any bounded function, we can obtain the error amplification of the entropy rate of $\widehat{\mathrm{Pr}}$ simply by choosing $f = -\log \widehat{\mathrm{Pr}}$.

Before we proceed, in order to rule out amplification of this entropy rate due to arbitrarily small probabilities (which can blow up $-\log \widehat{\mathrm{Pr}}$), it is helpful to define the $\gamma$-mixture distribution as: $\mathcal{D}^{(\gamma)} := (1 - \gamma)\mathcal{D} + \gamma \mathrm{Uni}$, where the $\mathrm{Uni}$ is the uniform distribution over all $M^T$ sequences. We will then consider the model $\widehat{\mathrm{Pr}}^{(\varepsilon)}$, which has only a minor degradation in the cross entropy compared to $\widehat{\mathrm{Pr}}$, and, yet, may have a large amplification in the entropy rate.

**Corollary 4.2.** *(Entropy rate amplification under generations) Suppose the bound in equation 1 holds. The $\varepsilon$-mixture distribution has KL bounded as:*

$$\frac{1}{T}\mathrm{KL}(\mathrm{Pr} \parallel \widehat{\mathrm{Pr}}^{(\varepsilon)}) \le \left(1 + \frac{2}{T}\right)\varepsilon.$$

*We have that:*

$$|\mathrm{CE}(\mathrm{Pr} \parallel \widehat{\mathrm{Pr}}^{(\varepsilon)}) - \mathrm{EntRate}(\mathrm{Pr})| \le \left(1 + \frac{2}{T}\right)\varepsilon, \text{ and}$$

$$|\mathrm{CE}(\mathrm{Pr} \parallel \widehat{\mathrm{Pr}}^{(\varepsilon)}) - \mathrm{EntRate}(\widehat{\mathrm{Pr}}^{(\varepsilon)})| \le \sqrt{2\varepsilon(T+1)}\left(\log M + \frac{\log(1/\varepsilon)}{T}\right).$$

This bound shows that, in the worst case, even a small cross entropy may provide little control over the generations under our model (in terms of entropy rate). In fact, for $\varepsilon = O(\frac{1}{T})$ (which we may hope is an accurate model), the bound is vacuous; a later remark shows this worst case bound is unimprovable, see the supplementary material.

The above theorems suggest that entropy rate amplification is a theoretical possibility in the worst case, which our experiments show is in fact prevalent in pratice. These entropy rate amplifications are evident from the plots in Figure 1. Regardless of the text corpus or the language model, we observe that the entropy rate under the model's generations quickly increases with time, indicating that this is a persistent problem even for state-of-the-art language models while generating text.

## 4.2 MODEL CALIBRATION

We now describe a procedure to fix this error amplification. First, let us define a distribution $\widehat{\mathrm{Pr}}_\alpha$ such that:

$$\widehat{\mathrm{Pr}}_\alpha(w_{1:T}) = \frac{\exp(\alpha f(w_{1:T})) \cdot \widehat{\mathrm{Pr}}(w_{1:T})}{Z_\alpha} \text{ where } Z_\alpha = \sum_{w_{1:T}} \exp(\alpha f(w_{1:T})) \cdot \widehat{\mathrm{Pr}}(w_{1:T}).$$

We can then recover a *calibrated* model that does not suffer from error amplification in $f$:

---

**Algorithm 1** (Inefficient) Entropy Rate Calibration

---

1: Input: Model $\widehat{\mathrm{Pr}}^{(\varepsilon)}$.
2: Define a model class:
$$\widehat{\mathrm{Pr}}_\alpha(w_{1:T}) = \left(\widehat{\mathrm{Pr}}(w_{1:T})^{(\varepsilon)}\right)^{1+\alpha} / Z_\alpha.$$

3: Fit $\alpha^*$: $\alpha^* = \mathrm{argmin}_\alpha \, \mathrm{CE}(\mathrm{Pr} \parallel \widehat{\mathrm{Pr}}_\alpha)$
4: Return $\widehat{\mathrm{Pr}}_{\alpha^*}$

---

**Lemma 4.3.** *(Calibration to $f$ with model improvement) Suppose the variance of $f$ is uniformly bounded in that there exists $\sigma_+^2$ such that the following holds for all $\alpha$, $\sigma_{\mathrm{Pr}_\alpha}^2(f) \leq \sigma_+^2$. Let $\alpha^* = \mathrm{argmin}_\alpha \, \mathrm{CE}(\mathrm{Pr} \parallel \widehat{\mathrm{Pr}}_\alpha)$. We have*

$$\mu_{\mathrm{Pr}}(f) - \mu_{\widehat{\mathrm{Pr}}_{\alpha^*}}(f) = 0, \;\; and \;\; \mathrm{CE}(\mathrm{Pr} \parallel \widehat{\mathrm{Pr}}_{\alpha^*}) \leq \mathrm{CE}(\mathrm{Pr} \parallel \widehat{\mathrm{Pr}}) - \frac{1}{T}\frac{(\mu(f) - \mu_{\widehat{\mathrm{Pr}}}(f))^2}{2\sigma_+^2}.$$

**Entropy rate calibration.** We can now apply the previous result to fix the entropy rate amplification seen in Table 1. Note that it is trivial to avoid the entropy rate amplification if we were allowed to degrade the quality of our model, in terms of perplexity (e.g. a unigram model does not have this amplification. However, we show that it is possible to match the entropy rate without having to sacrifice the quality of our model. In fact, we can both improve our model *and* more accurately match the entropy rate, by fitting a family of one-parameter models.

**Theorem 4.4.** *(Entropy rate calibration) Suppose equation 1 holds. Algorithm 1 returns a $\widehat{\mathrm{Pr}}_{\alpha^*}$ such that: the following calibration property is satisfied:*

$$\mathrm{CE}(\mathrm{Pr} \parallel \widehat{\mathrm{Pr}}_{\alpha^*}) = \mathrm{EntRate}(\widehat{\mathrm{Pr}}_{\alpha^*}).$$

*Furthermore, $\widehat{\mathrm{Pr}}_{\alpha^*}$ has entropy close to the true entropy rate as specified by:*

$$|\mathrm{EntRate}(\mathrm{Pr}) - \mathrm{EntRate}(\widehat{\mathrm{Pr}}_{\alpha^*})| \leq \left(1 + \frac{1}{T}\right)\varepsilon,$$

*and $\widehat{\mathrm{Pr}}_{\alpha^*}$ is an improvement over the original model as characterized by:*

$$\mathrm{CE}(\mathrm{Pr} \parallel \widehat{\mathrm{Pr}}_{\alpha^*}) \leq \mathrm{CE}(\mathrm{Pr} \parallel \widehat{\mathrm{Pr}}^{(\varepsilon)}) - \frac{1}{2}\left(\frac{\mathrm{CE}(\mathrm{Pr} \parallel \widehat{\mathrm{Pr}}^{(\varepsilon)}) - \mathrm{EntRate}(\widehat{\mathrm{Pr}})}{\log M + \frac{\log(1/\varepsilon)}{T}}\right)^2.$$

This result shows that we simply need a single parameter $\alpha$ to define a new model class that is a powered up version of our original model. Then, we can fit this $\alpha$ to minimize the cross-entropy of the new model with respect to the true distribution Pr, in order to eliminate the entropy rate amplification.

Even though this algorithm fits only a single parameter, it is not easily implementable since it requires an integration over sequences, at least in its exact form. One future direction would be to a sample based approach. This may be an interesting alternative to ideas like beam search (Steinbiss et al., 1994; Ortmanns and Ney, 2000; Antoniol et al., 1995), which also aims to minimize a global cost function on sequences that is inconsistent with the token-level perplexity loss used to train the underlying generative model.

**Lookahead algorithms.** In order to sidestep the computational issues of Algorithm 1, we provide another simple approach based on what can be viewed as a "one-step" lookahead correction (Algorithm 2). Let $\widehat{W}_t$ be a random variable with conditional distribution $\widehat{\mathrm{Pr}}(\cdot|W_{<t})$. $H(\widehat{W}_{t+1}|w_{\leq t})$ denotes the entropy of this conditional distribution, i.e.

$$H(\widehat{W}_{t+1}|w_{\leq t}) = \mathop{\mathbb{E}}_{w_{t+1} \sim \widehat{\mathrm{Pr}}(\cdot|w_{\leq t})}\left[\log \frac{1}{\widehat{\mathrm{Pr}}(w_{t+1}|w_{\leq t})}\right].$$

---

**Algorithm 2** Local Entropy Rate Calibration

---

1: Input: Model $\widehat{\mathrm{Pr}}^{(\varepsilon)}$, where $\widehat{W}_t \sim \widehat{\mathrm{Pr}}^{(\varepsilon)}(\cdot|W_{<t})$.
2: Define a model class:
$$\widehat{\mathrm{Pr}}_\alpha(w_{1:T}) = \widehat{P}_\alpha(w_1)\widehat{P}_\alpha(w_2|w_1)\dots$$
where
$$\widehat{\mathrm{Pr}}_\alpha(w_t|w_{<t}) = \widehat{\mathrm{Pr}}(w_t|w_{<t}) \cdot \exp\left(-\alpha \cdot H(\widehat{W}_{t+1}|w_{\leq t})\right)/Z_\alpha.$$
3: Fit $\alpha^*$: $\alpha^* = \mathrm{argmin}_\alpha \, \mathrm{CE}(\mathrm{Pr} \parallel \widehat{\mathrm{Pr}}_\alpha)$
4: Return $\widehat{\mathrm{Pr}}_{\alpha^*}$

---

Note that $H(\widehat{W}_{t+1}|w_{\leq t})$ includes the word $w_t$, so we require computing the entropy at time $t+1$ when predicting $W_t$ using a learned model.

For a conditional distribution, $\mathcal{D}(W_{1:T})$, let us define:

$$\bar{\mu}_\mathcal{D} = \frac{1}{T}\sum_{t=1}^{T} \mathbb{E}_{w_{<t}\sim\mathrm{Pr}} \, \mathbb{E}_{w_t\sim\mathcal{D}(\cdot|w_{<t})}[H(\widehat{W}_{t+1}|w_{\leq t})]$$

Thus, $\bar{\mu}_\mathcal{D}$ is the average of $H(\widehat{W}_{t+1}|w_{\leq t})$ with respect to a distribution which uses $\mathcal{D}$ for sampling the last word $W_t$ (at every timestep). Intuitively, the resulting model $\widehat{\mathrm{Pr}}_\alpha$ with a positive $\alpha$ would suppress sampling words leading to larger entropy but rather encourage words that stablizes the entropy 1-step ahead in the future. Therefore, if our learned language model $\widehat{\mathrm{Pr}}$ was accurate, we would hope that: $\bar{\mu}_{\mathrm{Pr}} \approx \bar{\mu}_{\widehat{\mathrm{Pr}}}$. The following corollary shows that this is achievable, along with improving the model's perplexity.

**Corollary 4.5.** *Suppose Equation 1 holds. Then, Algorithm 2 returns a $\widehat{\mathrm{Pr}}_{\alpha^*}$ such that:*

$$\bar{\mu}_{\mathrm{Pr}} - \bar{\mu}_{\widehat{\mathrm{Pr}}_{\alpha^*}} = 0, \quad and \quad \mathrm{CE}(\mathrm{Pr} \parallel \widehat{\mathrm{Pr}}_{\alpha^*}) \leq \mathrm{CE}(\mathrm{Pr} \parallel \widehat{\mathrm{Pr}}^{(\varepsilon)}) - \frac{1}{2}\left(\frac{\bar{\mu} - \bar{\mu}_{\widehat{\mathrm{Pr}}^{(\varepsilon)}}}{\log M + \frac{\log(1/\varepsilon)}{T}}\right)^2.$$

This result provides us with Algorithm 2, which is computationally quite tractable. We first use the learned model $\widehat{\mathrm{Pr}}$ to define a new model class $\widehat{\mathrm{Pr}}_\alpha$, which scales $\widehat{\mathrm{Pr}}$ by an exponential distribution over the weighted 1-step lookahead entropy $H(\widehat{W}_{t+1}|w_{\leq t})$. Then, similar to Algorithm 1, we simply fit the single parameter $\alpha$ to minimize the cross-entropy of the new model with respect to $\mathrm{Pr}$, which fixes the entropy amplification in the resulting model $\widehat{\mathrm{Pr}}_\alpha$. We observe this empirically in Figure 2 – our calibration results in a perplexity drop of almost 20 points over long-term generations under an LSTM model. Model and implementation details are in the supplementary material.

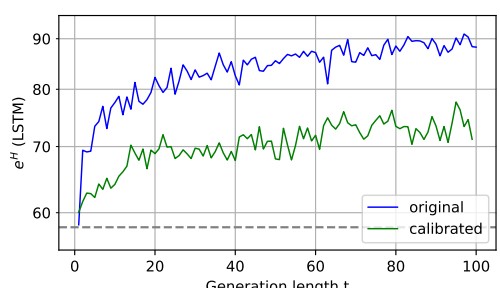

Figure 2: Effect of calibrating an LSTM generative model with 1-step lookahead. Blue: perplexity curve (i.e. exponential of conditional entropy $H$) from the setting of Figure 1. Green: the same perplexity measurements after applying local calibration.

**Generations from a calibrated model.** Table 2 provides sample generations from a calibrated Transformer model trained on the GBW dataset, compared to its original version. Qualitatively, the calibrated generations: (1) are shorter and more concise, and (2) display a better grasp of discourse structure across sentences. More generations are provided in the supplementary material.

## 5 CALIBRATION AND MEMORY

Defining a notion of memory in language models is challenging, and multiple equally sensible notions may co-exist. Here we present our choice from first principles. Let us say that $\widehat{W}_t$ is a sample

| Original model | Calibrated model |
|---|---|
| *Actual results could differ materially from those indicated by* these forward-looking statements as a result of various important factors , including , without limitation : changes in general economic and business conditions , including more difficult real estate environments ; **[...174 tokens...]** risks related to investigations by other companies ; inadequate information systems ; the impact of reduced availability of ; * assumptions upon such companies using such as ours to gauge CNET 's financial condition ; and other factors . | *Actual results could differ materially from those indicated by* these forward-looking statements as a result of a variety of factors , including but not limited to ( i ) the risk that the tender offer could close in one or more manner or at all ; ( ii ) risks associated with conducting business in foreign jurisdictions ; ( iii ) difficulties in combining some or all of the businesses under one roof ; ( iv ) decreased demand for electricity , natural gas and other energy products , including adverse effects on the pricing of oil and natural gas ; and ( v ) the risks associated with doing business internationally . |
| *Bluepoint Games , Inc. is a highly experienced* and multi-faceted publisher of licensed virtual worlds for gamers , developers and technology professionals . **[...114 tokens...]** James Upon , CEO of MyNetSheltetWeb and the three previous Developers of MySQL . Based in Redwood City , California , BlueMountain is the leader in franchise and game development for the massively multiplayer online game . | *Bluepoint Games , Inc. is a highly experienced* licensing , gaming and entertainment firm focused on developing the next generation of casual games based on the PlayStation ( R ) BRAVIA family of video game machines for the North American market . Bluepoint is a wholly owned subsidiary of Bluehill ID Holdings L.P. |

Table 2: Sample generations from a calibrated, state-of-the-art Transformer model trained on the GBW corpus, seeded with prefixes of sentences (in italics) from the holdout validation set.

from a model at time $t$, i.e. $\widehat{W}_t \sim \widehat{\Pr}(W_t | W_{<t})$. Let us also assume that $W_{<t} \sim \Pr(W_{<t})$. We will define the memory at gap $\tau$ as the mutual information between $\widehat{W}_t$ and the distant past (those words greater than $\tau$ steps ago) conditioned on the subsequence $W_{t-\tau:t-1}$. Precisely,

$$I_\tau := I(\widehat{W}_t ; W_{<t-\tau} | W_{t-\tau:t-1}) = H(\widehat{W}_t | W_{t-\tau:t-1}) - H(\widehat{W}_t | W_{<t}),$$

where we are not explicitly denoting the $t$ dependence in this definition[2].

Intuitively, $I_t$ can be viewed as how much uncertainty (entropy) in the prediction $W_t$ the model is able to reduce by utilizing the deep past $W_{<t-\tau}$ in addition to the recent past $W_{t-\tau:t-1}$.

The difficulty in estimating this mutual information is due to estimating $H(\widehat{W}_t | W_{t-\tau:t-1})$, which requires the marginalized model $\widehat{\Pr}(W_t | W_{t-\tau:t-1})$. To (even approximately) marginalize a model distribution $\widehat{\Pr}(W_t | W_{<t})$ over the deep past $W_{<t-\tau}$ is statistically difficult, since it requires the access to a pool of samples of $W_{<t}$ that share an *common* recent past $W_{t-\tau:t-1}$. Nevertheless, we now show that it is possible to obtain an upper bound (which is computationally efficient to estimate).

**Upper bounding mutual information using calibrated models.** In the above, we were considering the mutual information between $\widehat{W}_t$ and $W_{<t-\tau}$ conditioned on $W_{t-\tau:t-1}$. Let us now consider a more general setting, where we have a distribution $\Pr(Z, Y, X)$ where $Z$, $Y$, and $X$ are random variables. We wil eventually consider $Z, Y, X$ to be $\widehat{W}_t, W_{t-\tau:t-1} W_{<t-\tau}$, respectively.

For distributions $\mathcal{D}(\cdot | Y, X)$ and $\widetilde{\mathcal{D}}(\cdot | Y, X)$ and for $\alpha \in \mathbb{R}$, define

$$\mathcal{D}_\alpha(Z | Y, X) := \mathcal{D}(Z | Y, X) \cdot \left( \widetilde{\mathcal{D}}(Z | Y, X) \right)^\alpha / Z_\alpha .$$

We say that $\mathcal{D}(\cdot | Y, X)$ is calibrated to $\tilde{\mathcal{D}}(\cdot | Y, X)$, if $\mathcal{D} = \mathcal{D}_{\alpha=0}$ is unimprovable in that for all $\alpha$

$$\mathrm{CE}(\Pr \parallel \mathcal{D}) \leq \mathrm{CE}(\Pr \parallel \mathcal{D}_\alpha) .$$

Note this condition is achievable due to that calibrating a model to $\widetilde{\mathcal{D}}(\cdot | Y, X)$ involves a one dimensional (convex) estimation problem (over $\alpha$).

---

[2]While we may attempt to estimate $I_\tau$ for a given $t$, we can remove the $t$ dependence by either defining this quantity by with an average over $t$ or by using appropriate stationarity assumptions. In our experiments, we average over $t$.

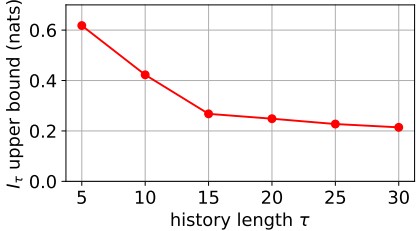

| $\tau$ | $\mathrm{CE}(\mathrm{Pr}\,\|\widetilde{\mathrm{Pr}})$ | $I_\tau$ upper bound | $\alpha^*$ |
|---|---|---|---|
| 5 | 4.8144 | 0.6180 | 0.003515 |
| 10 | 4.5258 | 0.4226 | -0.01041 |
| 15 | 4.4166 | 0.2678 | -0.00447 |
| 20 | 4.3347 | 0.2485 | -0.02268 |
| 25 | 4.2777 | 0.2274 | -0.01814 |
| 30 | 4.2408 | 0.2143 | -0.02323 |

Figure 3: *Left:* Plot of the upper bound on $I_\tau$ derived from calibrated models. *Right:* The measurements of the upper bound on mutual information, the cross entropy of the limited memory model $\widetilde{\mathrm{Pr}}$ as well as the optimal calibration coefficient $\alpha^*$ for various time lengths $\tau$. Details of the model used here can be found in the supplementary material.

**Theorem 5.1.** *Suppose we have a model $\widehat{\mathrm{Pr}}(Z|X)$, and suppose $\widetilde{Z} \sim \widetilde{\mathrm{Pr}}(\cdot|X)$, where $\widetilde{Z}$ is dependent only on $X$. Suppose that $\widehat{\mathrm{Pr}}$ is calibrated to $\widetilde{\mathrm{Pr}}$. Then we have that:*

$$I(\widehat{Z}; X|Y) \leq \mathrm{CE}(\mathrm{Pr} \,\|\, \widetilde{\mathrm{Pr}}) - H(\widehat{Z}|Y, X) \text{ , where:}$$

$$\mathrm{CE}(\mathrm{Pr} \,\|\, \widetilde{\mathrm{Pr}}) = \underset{Y \sim \mathrm{Pr}}{\mathbb{E}} \underset{Z \sim \mathrm{Pr}(\cdot|Y)}{\mathbb{E}} \left[ \log \frac{1}{\widetilde{\mathrm{Pr}}(Z|Y)} \right] .$$

**Memory estimation.** We first learn another $\widetilde{W}_t \sim \widetilde{\mathrm{Pr}}(\cdot|W_{t-\tau:t-1})$, and then calibrate $\widehat{\mathrm{Pr}}$ to $\widetilde{\mathrm{Pr}}$.

**Corollary 5.2.** *Suppose $\widehat{\mathrm{Pr}}^{\mathrm{cal}}(\cdot|W_{<t})$ is a model calibrated to $\widetilde{\mathrm{Pr}}(\cdot|W_{t-\tau:t-1})$. For a random variable, $\widehat{W}_t^{\mathrm{cal}} \sim \widehat{\mathrm{Pr}}^{\mathrm{cal}}(\cdot|W_{<t})$, we have that:*

$$I(\widehat{W}_t^{\mathrm{cal}}; W_{<t-\tau}|W_{t-\tau:t-1}) \leq \mathrm{CE}(\mathrm{Pr} \,\|\, \widetilde{\mathrm{Pr}}) - H(\widehat{W}_t^{\mathrm{cal}}|W_{<t}), \text{ where:}$$

$$\mathrm{CE}(\mathrm{Pr} \,\|\, \widetilde{\mathrm{Pr}}) = \underset{W_{t-\tau:t} \sim \mathrm{Pr}}{\mathbb{E}} \left[ \log \frac{1}{\widetilde{\mathrm{Pr}}(W_t|W_{t-\tau:t-1})} \right] .$$

This corollary gives us a means to efficiently provide upper bounds on the mutual information. The key is that since $\widetilde{\mathrm{Pr}}$ is efficiently computable, we can directly estimate $\mathrm{CE}(\mathrm{Pr}\,\|\widetilde{\mathrm{Pr}})$ through Monte Carlo estimation. We measure the upper bounds on $I_\tau$ of a LSTM model with trained limited-memory models $\widetilde{\mathrm{Pr}}$ (see details in the supplementary material) and report them in Figure 3. As expected, the memory estimate gradually decays with longer $\tau$, indicating that the models make more use of the recent past to generate text.

## 6 CONCLUSION

We have introduced a calibration-based approach to detect and provably correct the discrepancies between the long-term generations of language models and the true distributions they estimate sequentially. In particular, for state-of-the-art neural language models, we have observed large degradations of the entropy rate under iterative generation, and a proposed first-order correction which is both computationally tractable and effective. Using the same calibration approach, we have derived estimators for the amount of information extracted by these models from the deep past.

Aside from the empirical findings and improvements, we hope that this work will inspire a more principled line of discourse on the quality of long-term generations in language models. It remains an interesting open problem to relate the plethora of "future-aware" generation-improving heuristics to our calibration framework.

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

## A    PROOFS FOR SECTION 4

*Proof.* (of Lemma 4.1) We have

$$
\begin{aligned}
\left| \operatorname*{\mathbb{E}}_{w_{1:T}\sim\mathrm{Pr}}[f(w_{1:T})] - \operatorname*{\mathbb{E}}_{w_{1:T}\sim\widehat{\mathrm{Pr}}}[f(w_{1:T})] \right| &= \left| \sum_{w_{1:T}} \Big( \mathrm{Pr}(w_{1:T}) - \widehat{\mathrm{Pr}}(w_{1:T}) \Big) f(w_{1:T}) \right| \\
&\leq \left| \mathrm{Pr} - \widehat{\mathrm{Pr}} \right|_1 B \\
&\leq \sqrt{2\mathrm{KL}(\mathrm{Pr}\,||\widehat{\mathrm{Pr}})} B
\end{aligned}
$$

where we have used Holder's and Pinsker's inequalities Cover and Thomas (2006). □

*Proof.* (of Corollary 4.2) First observe:

$$
\log \frac{1}{\widehat{\mathrm{Pr}}^{(\varepsilon)}(w_{1:T})} \leq \log \frac{1}{(1-\varepsilon)\widehat{\mathrm{Pr}}(w_{1:T})} = \log \frac{1}{\widehat{\mathrm{Pr}}(w_{1:T})} - \log(1-\varepsilon) \leq \log \frac{1}{\widehat{\mathrm{Pr}}(w_{1:T})} + 2\varepsilon
$$

and that:

$$
\log \frac{1}{\widehat{\mathrm{Pr}}^{(\varepsilon)}(w_{1:T})} \leq \log \frac{M^T}{\varepsilon}. \tag{2}
$$

For the first claim, we have

$$
\frac{1}{T}\mathrm{KL}(\mathrm{Pr}\,||\widehat{\mathrm{Pr}}^{(\varepsilon)}) = \frac{1}{T} \operatorname*{\mathbb{E}}_{w_{1:T}\sim\mathrm{Pr}} \left[ \log \frac{\mathrm{Pr}(w_{1:T})}{\widehat{\mathrm{Pr}}^{(\varepsilon)}(w_{1:T})} \right] \leq (1 + \frac{2}{T})\varepsilon.
$$

using our assumption in Equation 1.

For the second claim, taking $f = \log \frac{1}{\widehat{\mathrm{Pr}}^{(\varepsilon)}(w_{1:T})}$ with Lemma 4.1, we have:

$$
\begin{aligned}
\left| \mathrm{CE}(\mathrm{Pr}\,||\widehat{\mathrm{Pr}}^{(\varepsilon)}) - \mathrm{EntRate}(\widehat{\mathrm{Pr}}^{(\varepsilon)}) \right| &\leq \frac{1}{T}\sqrt{2\mathrm{KL}(\mathrm{Pr}\,||\widehat{\mathrm{Pr}}^{(\varepsilon)})} \left| \log \frac{1}{\widehat{\mathrm{Pr}}^{(\varepsilon)}} \right|_\infty \\
&= \frac{1}{T}\sqrt{2\varepsilon(T+1)} \log \frac{M^T}{\varepsilon},
\end{aligned}
$$

which completes the proof. □

*Proof.* (of Lemma 4.3) By definition,

$$
\mathrm{CE}(\mathrm{Pr}\,||\widehat{\mathrm{Pr}}_\alpha) := \mathrm{CE}(\mathrm{Pr}\,||\widehat{\mathrm{Pr}}) - \frac{\alpha}{T} \operatorname*{\mathbb{E}}_{w_{1:T}\sim\mathrm{Pr}}[f(w_{1:T})] + \frac{1}{T}\log(Z_\alpha),
$$

we have:

$$
\frac{\partial \mathrm{CE}(\mathrm{Pr}\,||\widehat{\mathrm{Pr}}_\alpha)}{\partial \alpha} = \frac{1}{T}\left( -\mu_{\mathrm{Pr}}(f) + \mu_{\widehat{\mathrm{Pr}}_\alpha}(f) \right)
$$

The first claim now follows from optimality of $\alpha^*$.

For the second claim,

$$
\begin{aligned}
\frac{\partial^2 \mathrm{CE}(\mathrm{Pr}\,||\widehat{\mathrm{Pr}}_\alpha)}{\partial^2 \alpha} &= \frac{1}{T}\frac{\partial^2 \log(Z_\alpha)}{\partial^2 \alpha} \\
&= \frac{1}{T}\frac{\partial}{\partial \alpha} \frac{\sum_{w_{1:T}} f(w_{1:T}) \exp(\alpha f(w_{1:T})) \cdot \widehat{\mathrm{Pr}}(w_{1:T})}{\sum_{w_{1:T}} \exp(\alpha f(w_{1:T})) \cdot \widehat{\mathrm{Pr}}(w_{1:T})} \\
&= \frac{1}{T}\sigma^2_{\widehat{\mathrm{Pr}}_\alpha}(f) \leq \frac{\sigma^2_+}{T}.
\end{aligned}
$$

By Taylor's theorem, we have:

$$
\mathrm{CE}(\mathrm{Pr}\,||\widehat{\mathrm{Pr}}_\alpha) \leq \mathrm{CE}(\mathrm{Pr}\,||\widehat{\mathrm{Pr}}) - \alpha \cdot \frac{1}{T}\left( \mu_{\mathrm{Pr}}(f) - \mu_{\widehat{\mathrm{Pr}}}(f) \right) + \frac{\alpha^2}{2} \cdot \frac{\sigma^2_+}{T}.
$$

Taking the the $\alpha$ which minimizes the upper bound, leads to the second claim. □

**Remark A.1.** *(Sharpness) If $\varepsilon \geq \frac{1}{T}$, then there exists a problem where the bound is sharp and* $\mathrm{EntRate}(\widehat{\mathrm{Pr}})$ *takes on the maximal value of* $O(\log M)$. *As an example, consider a model $\widehat{\mathrm{Pr}}$, that starts by generating words under the true distribution* $\mathrm{Pr}$ *and has a $\frac{1}{T}$ probability of transitioning into a mode in which it generates words uniformly at random thereafter.*

*Proof.* (of Theorem 4.4) We can apply the previous lemma using

$$f = \log \frac{1}{\widehat{\mathrm{Pr}}(w_{1:T})},$$

and so our calibration condition implies:

$$0 = \mu_{\mathrm{Pr}}(f) - \mu_{\widehat{\mathrm{Pr}}_{\alpha^*}}(f) = -\left( \mu_{\mathrm{Pr}}(\log \widehat{\mathrm{Pr}}) - \mu_{\widehat{\mathrm{Pr}}_{\alpha^*}}(\log \widehat{\mathrm{Pr}}) \right).$$

Now observe that:

$$T \cdot \mathrm{CE}(\mathrm{Pr} \,\|\widehat{\mathrm{Pr}}_{\alpha^*}) = \mu_{\mathrm{Pr}}(-(1 + \alpha^*) \log \widehat{\mathrm{Pr}} + \log Z_{\alpha^*}) = -(1 + \alpha^*)\mu_{\mathrm{Pr}}(\log \widehat{\mathrm{Pr}}) + \log Z_{\alpha^*}$$

and, similarly,

$$T \cdot \mathrm{EntRate}(\widehat{\mathrm{Pr}}_{\alpha^*}) = -(1 + \alpha^*)\mu_{\widehat{\mathrm{Pr}}_{\alpha^*}}(\log \widehat{\mathrm{Pr}}) + \log Z_{\alpha^*}.$$

These imply:

$$\mathrm{CE}(\mathrm{Pr} \,\|\widehat{\mathrm{Pr}}_{\alpha^*}) - \mathrm{EntRate}(\widehat{\mathrm{Pr}}_{\alpha^*}) = -\frac{1}{T}(1 + \alpha^*)\left( \mu_{\mathrm{Pr}}(\log \widehat{\mathrm{Pr}}) - \mu_{\widehat{\mathrm{Pr}}_{\alpha^*}}(\log \widehat{\mathrm{Pr}}) \right) = 0,$$

which completes the proof of the first claim.

The proof of the second claim uses

$$\mu(f) - \mu_{\widehat{\mathrm{Pr}}}(f) = T\left( \mathrm{CE}(\mathrm{Pr} \,\|\widehat{\mathrm{Pr}}) - \mathrm{EntRate}(\widehat{\mathrm{Pr}}) \right),$$

and, by Equation 2,

$$\sigma_+^2 \leq T \log M + \log(1/\varepsilon),$$

which completes the proof. $\qquad\square$

Now we move on to the proof of Corollary 4.5.

Suppose $f(W_{\leq t})$ be a function of $W_{\leq t}$. For a conditional distribution, $\mathcal{D}(W_{1:T})$, let us now define:

$$\bar{\mu}_{\mathcal{D}}(f) = \frac{1}{T} \sum_{t=1}^{T} \mathop{\mathbb{E}}_{w_{<t} \sim \mathrm{Pr}} \mathop{\mathbb{E}}_{w_t \sim \widehat{\mathcal{D}}(\cdot|w_{<t})} [f(w_{\leq t})].$$

Define:

$$\widehat{P}_{t,\alpha}(w_t|w_{<t}) := \frac{1}{Z_{\alpha,t}} \exp(\alpha f(w_{\leq t})) \cdot \widehat{\mathrm{Pr}}(w_t|w_{<t})$$

and

$$\widehat{\mathrm{Pr}}_{\alpha}(w_{1:T}) := \widehat{P}_{1,\alpha}(w_1)\widehat{P}_{2,\alpha}(w_2|w_1)\ldots.$$

**Lemma A.1.** *Suppose $f \leq \sigma_+^2$. Let*

$$\alpha^* = \mathop{\mathrm{argmin}}_{\alpha} \mathrm{CE}(\mathrm{Pr} \,\|\widehat{\mathrm{Pr}}_{\alpha}).$$

*We have that:*

$$\bar{\mu}_{\mathrm{Pr}}(f) - \bar{\mu}_{\widehat{\mathrm{Pr}}_{\alpha^*}}(f) = 0$$

*and that*

$$\mathrm{CE}(\mathrm{Pr} \,\|\widehat{\mathrm{Pr}}_{\alpha^*}) \leq \mathrm{CE}(\mathrm{Pr} \,\|\widehat{\mathrm{Pr}}) - \frac{(\bar{\mu}(f) - \bar{\mu}_{\widehat{\mathrm{Pr}}}(f))^2}{\sigma_*^2}.$$

*Proof.* (sketch) The proof is identical to that of Lemma 4.3, with the addition of using linearity of expectation. $\qquad\square$

## B    PROOFS FOR SECTION 5

*Proof.* (of Theorem 5.1) It is convenient to define the distribution:

$$\mathcal{D}(Z, Y, X) = \widehat{\mathrm{Pr}}(Z|X, Y) \cdot \mathrm{Pr}(Y, X).$$

We then have:

$$I(\widehat{Z}; X|Y) = H(\widehat{Z}|Y) - H(\widehat{Z}|Y, X)$$

by the defintion of the mutual information.

The proof consists of showing that:

$$H(\widehat{Z}|Y) = E_{Y,Z\sim\mathcal{D}} \log \frac{1}{\mathcal{D}(Z|Y)} \leq \mathrm{CE}(\mathrm{Pr}\,||\widetilde{\mathrm{Pr}}).$$

Let us take $\widehat{\mathrm{Pr}}_\alpha(Z|X, Y) = \widehat{\mathrm{Pr}}(Z|X, Y) \cdot \left(\widetilde{\mathrm{Pr}}(Z|X)\right)^\alpha /Z_\alpha$. The zero gradient condition for the optimality at $\alpha = 0$ implies:

$$
\begin{aligned}
0 &= \left.\frac{\partial \mathrm{CE}(\mathrm{Pr}\,||\widehat{\mathrm{Pr}}_\alpha)}{\partial \alpha}\right|_{\alpha=0} \\
&= E_{X,Y\sim\mathrm{Pr}}\left[-E_{Z\sim\mathrm{Pr}(\cdot|X,Y)} \log \widetilde{\mathrm{Pr}}(Z|Y) + E_{Z\sim\widehat{\mathrm{Pr}}(\cdot|X,Y)} \log \widetilde{\mathrm{Pr}}(Z|Y)\right] \\
&= -E_{Y\sim\mathrm{Pr}}[E_{Z\sim\mathrm{Pr}(\cdot|Y)} \log \widetilde{\mathrm{Pr}}(Z|Y) + E_{X,Y\sim\mathrm{Pr}}[E_{Z\sim\widehat{\mathrm{Pr}}(\cdot|X,Y)} \log \widetilde{\mathrm{Pr}}(Z|Y)] \\
&= \mathrm{CE}(\mathrm{Pr}\,||\widetilde{\mathrm{Pr}}) + E_{X,Y\sim\mathrm{Pr}}[E_{Z\sim\widehat{\mathrm{Pr}}(\cdot|X,Y)} \log \widetilde{\mathrm{Pr}}(Z|Y)].
\end{aligned}
$$

This implies:

$$
\begin{aligned}
\mathrm{CE}(\mathrm{Pr}\,||\widetilde{\mathrm{Pr}}) &= E_{X,Y\sim\mathrm{Pr}}[E_{Z\sim\widehat{\mathrm{Pr}}(\cdot|X,Y)} \log \frac{1}{\widetilde{\mathrm{Pr}}(Z|Y)}] \\
&= E_{X,Y,Z\sim\mathcal{D}} \log \frac{1}{\widetilde{\mathrm{Pr}}(Z|Y)} \\
&= E_{Y,Z\sim\mathcal{D}} \log \frac{1}{\widetilde{\mathrm{Pr}}(Z|Y)} \\
&\geq E_{Y,Z\sim\mathcal{D}} \log \frac{1}{\mathcal{D}(Z|Y)} \\
&= H(\widehat{Z}|Y),
\end{aligned}
$$

where the last step uses the definition of $\widehat{Z}$ and Jensen's inequality. $\qquad\square$

## C    EXPERIMENTAL DETAILS

In this section, we outline the experimental setups used to obtain the empirical results throughout the paper. For the calibration and memory experiments (Table 1 row 1, Figure 1 (left), Figures 2, 3), our base model is a 3-layer LSTM with with 400 embedding dimension and 1150 hidden nodes. We train it on the Penn Treebank (PTB) corpus Marcus et al. (1993), following the setup of Merity et al. (2017) and Merity et al. (2018) for 500 epochs using SGD with batch size 20 and BPTT length 70. The trained base model achieves 64.3 validation perplexity and 58.3 test perplexity.

The limited-memory models $\widetilde{\mathrm{Pr}}(\cdot|W_{t-\tau:t-1})$ used for the memory estimation in Section 5 share the same architecture as our base model while, during training, the hidden states is re-initialized after reading every $\tau$ tokens ($\tau$ takes value from $\{5, 15, \ldots, 30\}$).

Finally, for the entropy rate measurements of larger-scale state-of-the-art language models (Table 1 rows 2-4, Figure 1 (right)), we used the pretrained weights published alongside Jozefow-icz et al. (2016); Radford et al. (2019) for rows 2 and 4, while we trained the model using the `tensor2tensor` framework. The model for row 2 is an LSTM with CNN-embedded inputs,

trained on the Google Billion Words (GBW) corpus. The other two are Transformer Vaswani et al. (2017a) models trained on GBW (row 3), and an proprietary corpus derived from a web crawl (Web-Text; row 4). For GPT-2, since the authors have not published training or validation data, we used the text of several New York Times articles as a stand-in validation set; the cross entropy loss is comparable to that reported on the validation set. The entropy rate amplification plot in Figure 1 (bottom) corresponds to the setup from row 4.

To measure the conditional entropy after $t$ generations, we measured the empirical conditional entropy of the $t$-th word over $> 500$ independent generations, which were produced by the standard way of iteratively sampling from the next predicted conditional distribution, seeded with ground-truth text up to $> 100$ random points in the validation set. We used the entropy rate at $t = 700$ as a proxy for the asymptotic limit in Table 1.

## D ADDITIONAL GENERATION SAMPLES

In this section, to provide a better sense of the qualitative effect of calibration, we provide below some additional generations, seeded by 10-token prefixes of the holdout (validation) sentences from the Google Billion Words dataset. Here, we used the model we trained for row 3 of Table 1. To identify a failure mode for the uncalibrated model, we selected the seed prefixes which resulted in unusually long generations by the uncalibrated model.

| Original model | Calibrated model |
|---|---|
| Actual results could differ materially from those indicated by these forward-looking statements as a result of numerous factors including the risks associated with the timely and efficient completion and integration of the Temporary Liquidity Guarantee Department 's supervision into the commercial , open market , solar energy , energy efficiency , electric utility transmission , and water demands of residential and business customers , Comcast 's ability to successfully implement its business plan , timing of completion of the acquisition and the effectiveness of the efforts and strategies involved in the integration of Rhapsody , timing of regulatory and client approvals and availability of key enhancements . | Actual results could differ materially from those indicated by these forward-looking statements as a result of a variety of factors , including but not limited to ( i ) the risk that the tender offer could close in one or more manner or at all ; ( ii ) risks associated with conducting business in foreign jurisdictions ; ( iii ) difficulties in combining some or all of the businesses under one roof ; ( iv ) decreased demand for electricity , natural gas and other energy products , including adverse effects on the pricing of oil and natural gas ; and ( v ) the risks associated with doing business internationally . |

| Actual results could differ materially from those indicated by these forward-looking statements as a result of various important factors , including , without limitation : changes in general economic and business conditions , including more difficult real estate environments ; declines in information technology spending ; continued availability of capital and government regulations ; changes in general economic and business conditions ; the possibility that extended unemployment and healthcare policies may change , or may reduce access to quality care services ; failure to obtain adequate and affordable medications ; changes in certain CME / CE product mix ; disruption in CME credit markets ; uncertainty of the outcomes of regulatory investigations of companies in which the Company has an interest ; dependence on suppliers for most of its products ; consolidation among financial institutions ; ability to attract and retain skilled personnel ; changes in rapidly changing technology and regulatory environments ; arrogance and complacency among financial analysts ; the impact of competition ; inability to retain and motivate senior management ; difficulties in the integration of acquired businesses ; the effects of redundancy and loss of key employees ; litigation , including claims and the challenge of insurance practices ; uncertainties relating to litigation ; risks related to investigations by other companies ; inadequate information systems ; the impact of reduced availability of ; * assumptions upon such companies using such as ours to gauge CNET 's financial condition ; and other factors . | Actual results could differ materially from those indicated by such forward-looking statements as a result of various important factors , including those discussed in the company 's periodic reports that are filed with the Securities and Exchange Commission and available on the SEC 's website at www.sec.gov. |
|---|---|

| | |
|---|---|
| Actual results could differ materially from those indicated by such forward-looking statements as a result of a variety of factors , including our ability to improve our liquidity . Among these factors are changes in the general economy , changes in political and economic conditions , changes in interest rates , changes in technology and implementation of regulatory policies and legislation , the direction of interest rates and changes in the banking industry , changes in loan prepayment activity , changes in consumer preferences and consumer and business lending markets , legislation or public compliance with applicable laws and regulations and changes in the business or regulatory environment . We caution you that there are many uncertainties that could cause actual results to differ materially from those indicated in the forward-looking statements . Among them are the risk factors that could cause results to differ from those expressed in the forward-looking statements . These factors include , but are not limited to : general economic and business conditions , including the financial markets ; fluctuations in interest rates ; government regulation of the financial services industry and possible failures ; planning assumptions and estimates ; potential funding requirements ; unexpected changes in cost increases ( including goodwill impairment ) ; competition ; the potentially lengthy , protracted U.S. recession ; and migratory consumer and business conditions . | Actual results could differ materially from those indicated by these forward-looking statements as a result of various important factors , including those discussed in the " Risk Factors " section of the Company 's Annual Report on Form 10-K for the most recently ended fiscal year . |
| Bluepoint Games , Inc. is a highly experienced and multi-faceted publisher of licensed virtual worlds for gamers , developers and technology professionals . The company is based in Vancouver , Canada . BlueKai 's innovative games are distributed by Devices EA , LLC , and Club Penguin . BlueKai owns and is the exclusive licensor of Scrabulous . BluetoothQ Interactive Inc. has acquired JoShear-Swain Media , LLC , a premier developer and publisher of community based games for the handheld game device . For further information , please visit : www.netgear.com / ngcleveld . Sprint 's fantasy game publisher and Web doing business within the Entertainment Group is James Upon , CEO of MyNetSheltetWeb and the three previous Developers of MySQL . Based in Redwood City , California , BlueMountain is the leader in franchise and game development for the massively multiplayer online game . | Bluepoint Games , Inc. is a highly experienced gaming and entertainment company with several renowned blockbuster franchises including PC , GameHouse ( ( R ) ) GameHouse ( ( R ) ) , Heavenly Sword ( ( TM ) ) , EverQuest ( R ) , Untold Story ( TM ) and EverQuest ( R ) II . Through its wholly-owned subsidiary , Bluehill ID ( R ) , the Bluehill ID logo and tagline are registered trademarks of Bluehill ID Corporation and its subsidiaries in the U.S. and in other countries . |

| | |
|---|---|
| Bluepoint Games , Inc. is a highly experienced gaming , entertainment and mobile games company with a vertically integrated portfolio including : games ( TM ) , social network , mobile , casual games , MMORPG , production , distribution , and licensing including its flagship games , SUIT and TIMMERIX ( TM ) , as well as its award-winning gaming , basketball and entertainment network . In order to create a highly integrated , pure and socially responsible Game ( R ) family , Bluepoint has collaborated with Amplify Systems International , Inc. on various titles for PlayStation ( R ) 2 , PLAYSTATION 3 ( R ) 5 , Wii ( TM ) 3 , PS3 , Wii ( TM ) ( and PS3 titles ) as well as PC games for PC , PSP , POOL , Wii ( TM ) ( and successor title ) and IP ( R ) , in addition to its focused gaming , entertainment and communication services . BlueBay 's exclusive licensee worldwide licensee of the Bluepoint ( TM ) ZMFAO Gateway series , it is the world 's leading portable gaming , PC and mobile phone company . For more information , see UNK , Inc. and '' Oakpoint : ZWC 's Community Health Business Development Center . | Bluepoint Games , Inc. is a highly experienced licensing , gaming and entertainment firm focused on developing the next generation of casual games based on the PlayStation ( R ) BRAVIA family of video game machines for the North American market . Bluepoint is a wholly owned subsidiary of Bluehill ID Holdings L.P. |
| Bluepoint Games , Inc. is a highly experienced , innovative entertainment sports gaming company whose products and services are used by some of the most recognized and respected names in the world of gaming including : Pokemon , Macau ( Valve ) , Quattro , Super Smash Bros. , Good Neighbor Games , IGN Games , Vail Resorts , Kania ( Ocean Spray , Pemberton and Roatenham ) , PURE Holdings , TeenNick , National Amusements , SEGA Games , Cirrus ( Aircraft ) and www.netapool.com. | Bluepoint Games , Inc. is a highly experienced player in the growing genre of casual games for both casual and active gaming enthusiasts . Bluepoint is an early stage Company with a significant following among youth and adults in Europe and the United States with an impressive track record in global on-line gaming opportunities . |
| Nursing Homes : Genworth 's 2009 Cost of Care Survey , conducted by the Robert Wood Johnson Foundation and released today , reveals the extent to which members of the U.S. population adheres to practices recommended since 1995 , including : a rolling three-hour '' Python for Life '' that fell asleep from 11 p.m. to 2 a.m. , sleep time from 11 p.m. to 3 a.m. , spare time from 8 a.m. to 9 p.m. , and use of state-of-the art non-invasive technologies . A remodeling and refurbishment of hospital facilities is underway as the nation 's economy begins to gain momentum . Similar to the previous years , Thinking About Health - Hear how health plans are working to address various congressional proposals to advance best practices in patient care and provide greater accountability , advocacy and transparency to consumers . | Nursing Homes : Genworth 's 2009 Cost of Care Survey is based on interviews with 516 family , friends and neighbors of insured and self-employed people conducted from Jan . |

| | |
|---|---|
| Nursing Homes : Genworth 's 2009 Cost of Care Survey is based on a double-blind , randomized , double-blind , placebo-controlled survey which involved an assessment of the cost-effectiveness of healthcare associated with an adequate diet and regular physical activity compared to its managed-care counterparts . The margin of error for this survey is + / - 3.3 percentage points at the 95 percent level of confidence . | Nursing Homes : Genworth 's 2009 Cost of Care Survey , conducted by Harris Interactive , performed significantly worse than a control group of its peers who provided care but were not able to offer health care to their employees . |
| Nursing Homes : Genworth 's 2009 Cost of Care Survey , conducted by CareScout ( R ) and published in the April 2009 issue , evaluated findings from the 10-year , nearly 900,000-member Specialty Health Management Association 's more than 6,000 professionals living in the United States . | Nursing Homes : Genworth 's 2009 Cost of Care Survey includes a series of health and medical cost reports on more than 100 home medical equipment and related products , including more than 3.9 million units of durable medical equipment . IBC 's cost of more than $ 100 billion is a significant portion of Medicare spending on home health care . |

Table 3: More generations from a state-of-the-art Transformer model trained on GBW, seeded with prefixes of sentences from the holdout validation set.

