# OpenReview forum: "Calibration, Entropy Rates, and Memory in Language Models"
_ICLR.cc/2020/Conference — Reject_

### Official Review · AnonReviewer1 · 2019-10-22
**Official Blind Review #1**

**Rating:** 6

**Review:**

This paper highlights and studies the problem of "entropy rate drift" for language models: the entropy rate of the language generated by a trained model is much higher than the entropy of ground truth sequences and this discrepancy worsen with the length of generation. The authors interestingly claim that the well-known lack of coherence in long-term model generations is due to this entropy rate drift. Valuably, the entropy rate drift is characterized mathematically. The authors propose a calibration method and prove that their method can interestingly reduce *both* the entropy rate drift and the perplexity of a miscalibrated model, *even though* they assume a rather simplistic model of miscalibration (one that leaks a small amount of mass to all the sequences of a given length). The author quantitatively show that their calibration method reduces the entropy rate drift and qualitatively show that their generations "make sense" in the long-term. As an auxiliary result, they show that a similar calibration method can be used to quantify the past information used by the model by upper-bounding the mutual information between the current prediction and the long-term past, given the short-term past, e.g. I(W_t | W<{t-\tau} | W_{t-\tau:t-1}).

I really enjoyed reading this paper and was really intrigued by the author's solution. However, in the current form, this paper is below the bar of acceptance due to some weaknesses: (i) rather strong assumption for the main derivation; (ii) lack of clarity and computational complexity of the proposed algorithms; (iii) weak experimental results and missing hint to current models / applications / concurring models. I would be more than happy to increase my score if the authors could kindly respond to the points below.

1) About the assumptions: in your theory, you show that the proposed solution is guaranteed to improve a particular constrained form of miscalibrated model P^\epsilon, in which a epsilon uniform distribution over all possible sentences of length K is added to the model distribution. This seems a rather strong assumption to me and bumps up to increasing the probability of each word by a small amount for each per-step conditional distribution estimated by the model.
1.1) Could you elaborate on why intuitively this assumption is something that happens in current models like GPT-2 ? Do you have a way of quantifying whether this assumption reasonably holds ?

2) About complexity:
2.1) What’s roughly the computational complexity of Algorithm 2 ?
2.2) Do you need to compute H(W_{t+1}|w_{<= t}) for all possible 1-step continuations w_t ? That seems quite expensive to me in the case of a large vocab. (e.g. sample a word, run forward one step and compute future entropy).
2.3) If this is correct, how to address this issue ? If I am missing something, it would be good to add to the paper some explicit considerations about the technical feasibility of the algorithm.

3) The amazing thing to me is that you reduce entropy-rate drift (although wrt a particular miscalibrated model, cf. 1) by re-minimizing cross-entropy using the ground-truth corpus. If you could give intuition for this , it would be great and make a much stronger paper !
3.1) Could you explain why intuitively this works ?
3.2) If \alpha is positive, Algorithm 2 is basically penalizing the model for producing words that lead to higher entropy in the future. Why this correlates to less cross-entropy with the data distribution ?
3.3) Why would calibrating a model that may not conform to the assumption 1) in general prefer to have an \alpha > 0 ?

4) About the experiments:
4.1) Where are the perplexities of the calibrated models ?
4.2) Are the perplexity improving as the theory would suggest ?
4.3) Is there any small , systematic human study that you could perform apart from just showing few examples of generations ?
4.4) Could you report some quantitative metrics of the generations from your models and the baselines, for example as computed in previous papers (https://arxiv.org/abs/1801.07736, https://arxiv.org/abs/1908.04319)
4.5) How do your generations change if you use beam search or arg-max, top-k sampling ? Does your method also help in preventing repetitions ?

Minor:

- I think you are missing a minus in Algorithm 2 inside the exponential.

====

Updated review:

I thank the authors for their answer. I think that overall this paper provides interesting insight on a hard problem.
I am raising my score and strongly hope that the authors add the considerations made in their response in the main paper: warning about complexity, optimal \alphas for each model and details.


**Experience Assessment:**

I have published one or two papers in this area.

**Review Assessment: Checking Correctness Of Derivations And Theory:**

I assessed the sensibility of the derivations and theory.

**Review Assessment: Checking Correctness Of Experiments:**

I carefully checked the experiments.

**Review Assessment: Thoroughness In Paper Reading:**

I read the paper thoroughly.

---

> ### Author Response · Authors · 2019-11-15
> **Response to R1**
>
> Thanks for the thorough review and many good questions.
>
> @Epsilon (1): See response to Reviewer 2 (“role of label smoothing”). We will include a note on this in the manuscript, and measure the effective \eps for various models as a sanity check.
>
> @Time complexity (2): The reviewer is correct: to perform an inference, Algorithm 2 requires |vocab|+1 inferences on the original model. This can be prohibitively expensive when the model is huge. Two comments: we view the computational cost to be the smallest one could expect for such a strong guarantee (bootstrapping a one-step predictor into calibrated generations), and GPU batching is possible for these intermediate calculations, as we have done. Furthermore, top-k truncation is possible (as well as possibly other heuristics).
>
> @Intuitions (3): (3.1) The qualitative intuition is perhaps captured by “think before you speak”: produce all candidate long generations, and reweigh them so that the entropy curve on that time horizon looks flat. The perhaps surprising mathematical fact is that this is possible with a one-parameter reweighting based on the entropy. (3.2) By definition of \alpha^*. If no such \alpha^* decreased the KL, then \alpha^* would be 0, and 1-step calibration would already hold. (3.3) Indeed, there is no guarantee that \alpha is negative. For some simple models (say, a Markov model with long-term behavior tending towards an absorbing set), there could be downward entropy rate drift, which calibration also corrects.
>
> @Experiment details (4): The perplexities are all very close to the reported entropies (i.e. the KL part of the cross entropy decomposition is small); which is why we didn’t report them redundantly. However, we agree that this is good to include, and will revise with these numbers. The PTB LSTM perplexity does improve by ~0.1. The Transformers have smaller \alpha^*, so we could not see statistically significant improvements. As for points 4.3 through point 4.5, we believe these could be interesting once the heuristics of calibration are fully fleshed out, but are outside the scope of this work.
>
> @Sign in exponential: Fixed.

---

### Official Review · AnonReviewer2 · 2019-10-25
**Official Blind Review #2**

**Rating:** 3

**Review:**

This paper presents a calibration-based approach to measure long range discrepancies between a model distribution and the true distribution in terms of the difference between entropy rate and cross entropy, which is exactly the forward KL divergence. It propose a simple one parameter estimation to improve the model and provides experiments to show the effectiveness.

This paper in fact provides theoretical justification to the so-called temperature sweep method that is hotly debated in the area of text generation. Several issued should be clearly addressed before the acceptance for publication.

1. The authors should read the Language GANs falling short paper, and conduct the experiments in the same way in that paper and compare their approach with temperature sweep method. The temperature sweep method is only used at inference stage. The proposed approach, Algorithm 2, is used at training stage.

2. The paper provides theoretical results in terms of population distribution, the true but known distribution, however in practice, empirical distribution is used instead, for example the cross entropy in section 3 used in training, are these theoretical results still valid for empirical distribution? If yes, please state in the paper, if not please state why?

3. On page 6, first line, it is stated "Since this holds for any bounded function, we can obtain the error amplification of the entropy rateof \hat{Pr} simply by choosing f = − log \hat{Pr}." The log function is unbounded, so please be careful. Fortunately \hat{Pr}^{\epsilon} is bounded, so Corollary 4.2 is correct. For the proof of Corollary 4.2, I don't know how to get the inequality in (2) and the first claim, so please provides more steps or explanations.

4. How the entropy rate of each language model in Table 1 is obtained?

5. In section 3, capital letter is used for random variable, but to define H, CR, EntRate, KL, small letter is used, which is not consistent. Also some is used as subscript, some is under E

6. Many unconditional language model papers are not cited, for example, ELMO, BERT, XLNet, Albert et la. and many language GANs paper. On the other hand, many papers for conditional language models are cited,  these papers are not appropriate to cite since the paper targets on unconditional language model.

7. In the first paragraph of page 1, there is a statement of "Capturing long-termdependencies has especially been a major focus, with approaches ranging from explicit memorybased neural networks (Grave et al., 2016; Ke et al., 2018) to optimization improvements to stabilizelearning (Le et al., 2015; Trinh et al., 2018)."

In the second paragraph of page 1, there is a statement of "Capturing long-term dependencies has especially been a major focus, with approaches ranging from explicit memory-based neural networks (Grave et al., 2016; Ke et al., 2018) to optimizationimprovements aimed at stabilizing training (Le et al., 2015; Trinh et al., 2018)."

This is redundant.

8. This paper focuses  on the forward KL divergence, which is related to the quality of language model, but doesn't anything about diversity of language model, which is related to the reverse KL divergence? Can it be extended to the reverse KL divergence?

Missing references:

M. Caccia, L. Caccia, W. Fedus, H. Larochelle, J. Pineau, and L. Charlin. Language GANs falling short. In Neural Information Processing Systems Workshop on Critiquing and Correcting Trends in Machine Learning, 2018.

William Fedus, Ian J. Goodfellow, and Andrew M. Dai. MaskGAN: Better text generation via filling in the . ICLR, 2018.

Jiaxian Guo, Sidi Lu, Han Cai, Weinan Zhang, Yong Yu, and Jun Wang. Long text generation via adversarial training with leaked information. AAAI, 2018.

Ferenc Huszar. How (not) to train your generative model: Scheduled sampling, likelihood, adversary? arXiv preprint arXiv:1511.05101, 2015.

Lantao Yu, Weinan Zhang, Jun Wang, and Yong Yu. SeqGAN: Sequence generative adversarial nets with policy gradient. AAAI, 2017.

Zhongliang Li, Tian Xia, Xingyu Lou, Kaihe Xu, Shaojun Wang, and Jing Xiao. Adversarial discrete sequence generation without explicit neural networks as discriminators. AISTATS, 2019.

Kevin Lin, Dianqi Li, Xiaodong He, Zhengyou Zhang, and Ming-Ting Sun. Adversarial ranking for language generation. NIPS, 2017.

Ehsan Montahaei, Danial Alihosseini, and Mahdieh Soleymani Baghshah. Jointly measuring diversity and quality in text generation models. NAACL-HLT, 2019.

A Quality-Diversity Controllable GAN for Text Generation


**Experience Assessment:**

I have published in this field for several years.

**Review Assessment: Checking Correctness Of Derivations And Theory:**

I carefully checked the derivations and theory.

**Review Assessment: Checking Correctness Of Experiments:**

I carefully checked the experiments.

**Review Assessment: Thoroughness In Paper Reading:**

I read the paper thoroughly.

---

> ### Author Response · Authors · 2019-11-15
> **Response to R2**
>
> We are grateful for the thoughtful comments, and for the wide range of suggested references. We have already cited the mentioned MaskGAN paper.
>
> @Temperature sweep, Language GANs (1): Though our proposed method bears a superficial resemblance to softmax temperature tuning (which we mention and cite), it is *not* equivalent. The softmax temperature adjustment considered by that paper (and many others) is constant. Ours is determined by the per-word lookahead entropy, which could deviate arbitrarily from a constant adjustment.
>
> @Empirical vs. population Pr (2): All of our results hold if Pr is assumed to be the empirical distribution over a finite independent sample from the ground truth (thus, the unknown ground truth never needs to appear anywhere in our theoretical results). We have purposefully avoided analyzing the generalization gap. Formally, this appears in our assumption that we can fit the \alpha parameter on the population Pr.
>
> @Role of label smoothing (3): The intuition is that smoothing by \eps avoids unrecoverable degenerate cases, in which \hat{Pr} gets infinite cross entropy vs. the true distribution and cannot be calibrated. So, while it’s true that this is required for our results to hold, the purpose is to avoid these vacuous cases. We believe that in practical models, probability vectors are always bounded away from the boundary of the simplex.
>
> @Entropy rate estimate (4): For one step: like measuring the usual cross entropy on the evaluation set, but ignore the label and average the entropy of the predicted probability vector instead. For multiple steps: do the same, but feed the model $t$ of its own generations first, and average over many trajectories.
>
> @Notation (5): Throughout, we have used W to denote a random variable, and w to mean its possible values; we will include a note to clarify this.
>
> @Unconditional LM (6): We had decided not to mention BERT/etc. because these non-autoregressive LMs do not specify unique probability distributions over sequences; thus, our methods don't immediately apply to those models. See response to Reviewer 4. Nevertheless, since this has been mentioned by two reviewers, we have revised the manuscript to cite those works, along with a short discussion.
>
> @Typo (7): Fixed.
>
> @Reverse KL (8): This is an interesting direction, but our method only clearly has end-to-end guarantees when the language model is autoregressively factorized. It would require significant modification (and perhaps loss of theoretical guarantees) to obtain the same result for reverse KL.

---

### Official Review · AnonReviewer4 · 2019-11-04
**Official Blind Review #4**

**Rating:** 6

**Review:**

(emergency review)

This paper demonstrates that a left-to-right language model suffers a high entropy rate when generating a long-term sequence of words. Then the authors claim that this is because of entropy rate amplification, which could be mitigated by 'calibration'. With local entropy rate calibration, a language model could achieve lower perplexity generating shorter and concise sequences of words.

The proposed technique (local entropy rate calibration) is straightforward to implement, and empirically shown to be effective. This would be easily applied to the decoder in many seq2seq models, expected to improve various language generation tasks. However, other language models that use bi-directional connections (BERT, RoBERTa, ALBERT) or GAN based language generation models are omitted, and I think these models should be considered to make this work have more impact.



**Experience Assessment:**

I have read many papers in this area.

**Review Assessment: Checking Correctness Of Derivations And Theory:**

I assessed the sensibility of the derivations and theory.

**Review Assessment: Checking Correctness Of Experiments:**

I assessed the sensibility of the experiments.

**Review Assessment: Thoroughness In Paper Reading:**

I made a quick assessment of this paper.

---

> ### Author Response · Authors · 2019-11-15
> **Response to R4**
>
> Thanks for the encouraging review.
>
> @Bidirectional models: The trouble with applying our methods to these non-autoregressive language models is that there isn’t a single probabilistic model to improve. For example, under different masking patterns, there is no guarantee that BERT outputs conditional probabilities consistent with a single distribution over sequences. Indeed, it remains an open line of research to extract probabilistic models from these non-autoregressively-factorized models (see, e.g. [1]). We've revised the manuscript with a few citations and a small note about these.
>
> [1] BERT has a mouth, and it must speak: BERT as a markov random field language model. Wang & Cho ‘19.

---

### Author Response · Authors · 2019-11-15
**Comment to all reviewers about quantitative vs. qualitative objectives**

We thank all the reviewers for their input. A common thread among all of the reviewers’ responses is a lack of treatment/comparison to the plethora of existing text generation heuristics. We have responded about specific comparisons in the individual comments. However, we believe there has been one primary misconception:

We emphasize that our contribution is distinguished from all of the heuristics we (and the reviewers) have cited, in that it is a *provable* improvement on all *black-box* sequence models. In this sense, we view the main contribution as theoretical groundwork (identifying and correcting entropy rate drift), rather than developing a fully-fleshed-out heuristic tool for qualitative generation improvement. Bridging these goals is an exciting future direction, and we expect that it will require some careful heuristic modifications. Though we mention this in the intro, we will overhaul the exposition to make this point clearer.

For now, we've uploaded a minor revision to the manuscript, addressing the cosmetic issues pointed out by the reviewers.

---

### Decision · Program_Chairs · 2019-12-19

**Decision:**

Reject

**Comment:**

This paper shows empirically that the state-of-the-art language models have a problem of increasing entropy when generating long sequences. The paper then proposes a method to mitigate this problem. As the authors re-iterated through their rebuttal, this paper approaches this problem theoretically, rather than through a comprehensive set of empirical comparisons.

After discussions among the reviewers, this paper is not recommended to be accepted. Some skepticism and concerns remain as to whether the paper makes sufficiently clear and proven theoretical contributions.

We all appreciate the approach and potential of this paper and encourage the authors to re-submit a revision to a future related venue.